# Emergence of NDM-1-Producing *Pseudomonas aeruginosa* Nosocomial Isolates in Attica Region of Greece

**DOI:** 10.3390/microorganisms12091753

**Published:** 2024-08-23

**Authors:** Olga Pappa, Christina Louka, Kleon Karadimas, Evangelia Maikousi, Angeliki Tzoukmani, Michalis Polemis, Anna-Danai Panopoulou, Ioannis Daniil, Stella Chryssou, Kassiani Mellou, Jette S. Kjeldgaard, Olympia Zarkotou, Costas Papagiannitsis, Kyriaki Tryfinopoulou

**Affiliations:** 1AMR and HAIs Laboratory, Central Public Health Laboratory, National Public Health Organization, Vari, 16672 Attica, Greece; o.pappa@eody.gov.gr (O.P.); k.karadimas@eody.gov.gr (K.K.); euamaik@gmail.com (E.M.); aggeliki_tzoukmani@hotmail.com (A.T.);; 2Clinical Microbiology Laboratory, Tzaneion General Hospital of Piraeus, 18536 Attica, Greece; cslouka@gmail.com (C.L.); drdaniel.gr@gmail.com (I.D.); olyzar@hotmail.com (O.Z.); 3Clinical Microbiology Laboratory, Syggros Hospital of Athens, 16121 Attica, Greece; panopoulouad@gmail.com (A.-D.P.); chryssoustella@gmail.com (S.C.); 4Directorate of Epidemiological Surveillance and Response for Infectious Diseases, National Public Health Organization, 15123 Athens, Greece; k.mellou@eody.gov.gr; 5European Union Reference Laboratory for Antimicrobial Resistance (EURL-AR), National Food Institute, Technical University of Denmark, 2800 Kongens Lyngby, Denmark; jetk@food.dtu.dk; 6Department of Microbiology, University Hospital of Larissa, 41334 Larissa, Greece; c.papagiannitsis@gmail.com; 7Clinical Microbiology and Microbial Pathogenesis Laboratory, School of Medicine, University of Crete, Heraklion, 71500 Crete, Greece

**Keywords:** *P. aeruginosa*, multidrug resistance, high-risk clones, NDM-1, Greece

## Abstract

Here, we report on the emergence and spread of multidrug-resistant NDM-1-producing *P. aeruginosa* isolates from patients hospitalized in the Attica region, Greece, in 2022 to provide data on their resistome, their virulome, the genetic environment of *bla*_NDM-1_, and their molecular epidemiology. A total of 17 carbapenem-resistant *P. aeruginosa* isolates identified as NDM-producers by immunochromatography at the hospital level were sent to the Central Public Health Laboratory, in the frame of the laboratory surveillance of carbapenem-resistant pathogens, for further characterization. The initial screening for genetic AMR determinants was carried out by PCR and the MDR Direct Flow Chip assay. Typing was performed by MLST and DLST, the latter in a subset of isolates. Further analysis was performed by whole-genome sequencing (WGS) of six isolates from both hospitals to analyze their entire genomes and elucidate their genetic relatedness. All isolates were allocated to international high-risk clones, sixteen to ST773 and one to ST308. Five ST773 and the sole ST308 isolate were found to harbor the *bla*_NDM-1_ gene, along with various other ARGs integrated into their chromosomes, as well as with a wide variety of virulence genes. The *bla*_NDM-1_ gene was located in the integrative and conjugative elements ICE*6600*-like and ICE*_Tn4371_6385* in ST773 and ST308 isolates, respectively. Single-nucleotide polymorphism analysis of the five ST773 isolates indicated their clonal spread in both hospitals. These results suggested that two different molecular events contributed to the emergence of NDM-1-producing *P. aeruginosa* isolates in Athenian hospitals, highlighting the need for ongoing surveillance.

## 1. Introduction

*Pseudomonas aeruginosa* is a significant cause of nosocomial infections, particularly in critically ill and immunocompromised patients, and is associated with high morbidity and mortality worldwide [1]. The extensive complexity and plasticity of its genome, with a wide array of virulence genes and complex regulator and signaling systems, contribute to *P. aeruginosa* pathogenesis in both acute and chronic infections [1]. Accordingly, the alarming rates of multidrug resistance (MDR) in *P. aeruginosa* is a major public health problem [2]. Notably, the ability of *P. aeruginosa* to develop resistance through multiple mechanisms, including the selection of mutations in chromosomal genes and the accumulation of transferable resistance determinants, has resulted in extensively drug-resistant (XDR) clinical isolates [3].

In particular, carbapenem resistance in *P. aeruginosa* is commonly the result of a combination of the following mechanisms: (a) the overexpression of efflux pump systems [4], (b) the overexpression (via derepression) of chromosomal cephalosporinase [5], and (c) the decreased expression or loss of the outer membrane protein OprD [6]. In addition, carbapenem resistance among *P. aeruginosa* can result from the acquisition of acquired carbapenemases, mainly Ambler class B metallo-β-lactamases (MBLs) [7]. Notably, the recent multi-national ERACE-PA Surveillance program provided evidence of the increasing prevalence of carbapenemase producers among carbapenem-resistant *P. aeruginosa* (CRPA) isolates [8]. Of the 807 CRPA collected between 2019 and 2021 from 17 centers in 12 countries from America, Europe, Asia, and Africa, 33% tested carbapenemase-positive phenotypically, and of these, 86% were genotypically positive, with the most common being VIM, followed by GES [8].

In Greece, VIM-type MBLs have been the predominant enzymes in CRPA isolates since the late 1990s [9,10,11,12,13]. Regarding NDM-type MBLs, while NDM-1-producing *P. aeruginosa* isolates were detected for the first time in the Balkan area (Serbia) in 2010 [14], hospital outbreaks have been reported in the Balkan countries only recently [15,16,17,18], indicating a possible shift in the molecular epidemiology of CRPA in the area.

Notably, even though the *P. aeruginosa* population structure has generally been considered panmictic, including many different genotypes that frequently recombine, the nosocomial MDR and XDR *P. aeruginosa* strains are largely allocated to 10 main high-risk clones (sequence types ST111, ST175, ST233, ST235, ST244, ST277, ST298, ST308, ST357, and ST654). It is well established that these high-risk, international *P. aeruginosa* clones are characterized by their ability to acquire resistance determinants through horizontal gene transfer, especially horizontally acquired-β-lactamases such as extended-spectrum β-lactamases (ESBLs) and MBLs such as VIM, IMP, and NDM [1,19,20].

In order to track high-risk clones and to analyze the emergence of novel resistance determinants and their mode of transmission, along with their main virulence characteristics, continuous monitoring and laboratory surveillance programs are mandatory [20]. In Greece, since the implementation of the National Action Plan in 2010 for the prevention and control of nosocomial infections caused by carbapenem-resistant Gram-negative pathogens in healthcare settings [21], protocols for sending important isolates to reference laboratories, such as the Antimicrobial Resistance and Health Care-Associated Infections Laboratory (AMR and HAIs Lab) of the Central Public Health Laboratory (CPHL), for confirmation and further characterization have been implemented for laboratory surveillance purposes.

Here, we report on the emergence of multidrug-resistant NDM-1-producing *P. aeruginosa* isolates from clinical samples collected in two hospitals in the Attica region, Greece, in 2022, focusing on their resistome, their virulome, the genetic environment of *bla*_NDM-1_, and their molecular epidemiology.

## 2. Materials and Methods

### 2.1. Bacterial Collection

The bacterial collection consisted of 17 *P. aeruginosa* clinical isolates recovered during 2022 and identified as NDM-producers by immunochromatography (NG-Test Carba 5-Biotech, Guipry, France) in the clinical microbiology laboratories of two hospitals (H1 and H2) in the Attica region of Greece. Fifteen isolates from H1 were randomly selected among the 57 NDM-positive *P. aeruginosa* isolates found in the hospital from January to September 2022, while in H2, only two NDM-1-producing *P. aeruginosa* isolates were found in late 2022. All 17 isolates were referred to the reference AMR and HAIs laboratory of the CPHL for further characterization.

### 2.2. Antimicrobial Susceptibility Testing

The minimum inhibitory concentrations (MICs) of piperacillin, ceftazidime, cefepime, aztreonam, gentamicin, amikacin, ciprofloxacin, imipenem, and meropenem were determined using the VITEK 2 system (bioMerieux, Inc. Durham, NC 27712, USA), while the MIC of colistin was determined by broth microdilution. *P. aeruginosa* ATCC27853 and *Escherichia coli* ATCC 25922 were used as control strains.

### 2.3. Detection of Antimicrobial Resistance Genes and Known Chromosomal Point Mutations

*P. aeruginosa* genomic DNA was extracted using Easy Extraction Solutions, Primer Design (Eastleigh, UK), following the manufacturer’s instructions, after 48 h growth in nutrient broth and nutrient agar. All isolates were initially screened by PCR for various carbapenemase genes (*bla*_VIM_, *bla*_NDM_, *bla*_KPC_, and *bla*_OXA-48_) using previously described primers and conditions [22].Moreover, all isolates were tested using the MDR Direct Flow Chip (Máster Diagnóstica, Granada, Spain), a DNA microarray-based assay for antimicrobial resistance gene detection that includes 56 AMR determinants for β-lactams, quinolones, aminoglycosides, macrolides, sulfonamides, colistin, vancomycin, chloramphenicol, and linezolid. *P. aeruginosa* isolates were cultured overnight on Nutrient agar (Oxoid, Hampshire, UK). A single colony was homogenized in 50 mL of sterile distilled water, of which a volume of 5 mL was directly used for AMR molecular testing, without prior DNA extraction. The MDR Direct Flow Chip assay is based on multiplex PCR amplification, followed by automatic reverse hybridization using a membrane containing specific probes to detect 56 known resistance markers in the automatic hybriSot 12 platform [resistance markers: *mecA*, *mecC*, *vanA*, *vanB*, *bla*_SHV_, *ESBL SHV-type (bla*_SHV-S_, *bla*
_SHV-SK_*)*, *ESBL CTX-M-type (bla*_CTX_*)*, *β-Lactamase AMPC genes (bla*_DHA_, *bla*_CMY_*)*, *bla*_KPC_, *bla*_SME_, *bla*_NMC/IMI_, *bla*_GES_, *bla*_VIM_, *bla*_GIM_, *bla*_SPM_, *bla*_NDM_, *bla*_SIM_, *bla*_IMP-like_, *oxa48-like oxa23-like*, *oxa24-like*, *oxa51-like*, *oxa58-like*, *mcr1*, *mcr2*, *sul1*, *sul2*, *sul3*, *msrA*, *mef*, *ermA*, *ermB*, *ermC*, *aac*, *armA*, *rmtB*, *rmtC*, *rmtF*, the *Escherichia coli* gyrase A WT gene (*gyrA*) for the detection of *Escherichia coli* and mutations associated with resistance to fluoroquinolones (*gyrE*-S83L, *gyrE*-S83L-D87G, *gyrE*-S83L-D87N, *gyrE*-S83W-D87G), the *E. coli* topoisomerase IV gene *(parC*) for the detection of the quinolone resistance-associated mutation *parE*-S80I, and the *P. aeruginosa* gyrase A gene (*gyrA-Pae*) for the detection of fluoroquinolone resistance-associated mutations (*gyrP*-T83I, *gyrP*-T83I-D87N, *gyrP*-T83I-D87G), *qnrA*, *qnrB*, *qnrS*, *oqxA*, *oqxB*, *cfr*, *catB3*]. Positive signals are visualized through a colorimetric immunoenzymatic reaction in a chip membrane by the HS12 hybridization platform, which includes a built-in camera that captures an image of the chip and analyzes the dot pattern by means of the hybrisoft software (HSHS version 2.02.00.R09.01).

### 2.4. Molecular Typing

The *P. aeruginosa* isolates were typed below species level using Multi-Locus Sequence Typing [MLST] analysis, and in 4 of them, Double-Locus Sequence Typing (DLST) was also applied.

MLST was applied by amplifying seven housekeeping genes (*acsA*, *aroE*, *guaA*, *mutL*, *nuoD*, *ppsA*, and *trpE*) according to the protocol on the *P. aeruginosa* MLST website (https://pubmlst.org/organisms/pseudomonas-aeruginosa, accessed on 5 October 2022 and 25 January 2023). A sequence analysis of PCR amplicons using both methods was performed by Eurofins Genomics (https://eurofinsgenomics.eu/), The sequences and chromatographs were analyzed and interpreted by 4peaks software v. 1.8.

DLST was implemented in 4 *P. aeruginosa* representative isolates from both MLST types and both hospitals (9912, 9953, 9964, 9965; Table 1 and Appendix A). DLST sequences were compared to the DLST database (http://www.dlst.org/Paeruginosa/, accessed on 5 October 2022 and 25 January 2023) for allele assignment of the genetic markers ms172 and ms217; if there was no identification for the submitted locus, the procedure for the submission of new alleles in the DLST database was followed, and a new locus number was assigned [23].

### 2.5. Whole-Genome Sequencing (WGS)

WGS was performed in six isolates (9953, 9958, 9964, 9965, 9912, and 10071; Table 1). After determining DNA integrity and quality, DNA was prepared and sequenced at Eurofins Genomics (Constance, Germany) using an INVIEW Genome (INVIEW Resequencing Bacteria) product. This included fragmentation, end-repair and dA-tailing, adapter ligation, size selection, and library amplification. The prepared libraries were then quality-checked, pooled, and sequenced on an Illumina platform (Illumina NovaSeq6000, PE150 mode) (Illumina Inc., San Diego, CA, USA) with an average coverage of 280x. High-quality reads were de novo assembled by SPAdes v3.11.1.

In addition, Nanopore sequencing was performed at the CPHL for two isolates (9912 and 9965) in order to study the genetic environment of the mobile genetic elements (MGEs) implicated in the dissemination of *bla*_NDM_ and investigate for the presence of plasmids. Genomic DNA was extracted using the Monarch high-molecular-weight (HMW) DNA extraction kit for tissue (T3060L; New England Biolabs, Ipswich, MA, USA). For Nanopore sequencing, the total DNA was barcoded using the rapid barcoding kit (SQK-RBK004) and sequenced on an R10.4.1 flow cell with a MinION device. The reads were base-called with MinKNOW (5.9.18). The Nanopore reads were demultiplexed, and adapters were trimmed using Porechop 0.2.4. Flye 2.9.3 was used for de novo assembly using default settings and Medaka 1.7.2 for assembly polishing via neural networks. ABRicate 1.0.1 was used for the mass screening of contigs for antimicrobial resistance and virulence genes. AMRFinderPlus 3.12.8 (https://www.ncbi.nlm.nih.gov/pathogens/antimicrobial-resistance/AMRFinder/, accessed on 30 May 2023), the NCBI antimicrobial resistance gene and resistance-associated point mutation finder, was used for ARG and point mutations associated with antimicrobial resistance. The virulence factor database (VFDB) was used to classify virulence and stress resistance genes (http://www.mgc.ac.cn/VFs/, accessed on 26 July 2024). Finally, ICEfinder (https://bioinfo-mml.sjtu.edu.cn/ICEfinder/ICEfinder.html, accessed on 4 July 2024), PlasmidFinder (https://cge.food.dtu.dk/services/PlasmidFinder/, accessed on 4 July 2024), and Blast webtool (https://blast.ncbi.nlm.nih.gov/Blast.cgi, accessed on 4 July 2024) were used to identify horizontal transfer genetic elements

### 2.6. Phylogenomic Relationship

The selected genomes of six *P. aeruginosa* isolates were analyzed for phylogenomic relationships using CSI phylogeny, which is accessible from the Center for Genomic Epidemiology (www.genomicepidemiology.org, accessed on 29 August 2023). Single-nucleotide polymorphisms (SNPs) were determined using the following settings: the Z-score cutoff value was 1.96, the minimal depth at SNP positions was 10, and SNPs within 100 base pairs were removed (pruning). The SNP count matrix from CSI phylogeny was generated using the internal isolate 10071 assembly or the previously described South Korean blaNDM-1 *P. aeruginosa* ST773 (CP053917) as the reference.

### 2.7. Nucleotide Accession Numbers

The genomes of the six *bla*_NDM-1_-positive *P. aeruginosa* isolates have been deposited in ENA under submission account ID Webin-64929 and sample ID ERS15538483-88.

## 3. Results

### 3.1. Bacterial Isolates and Details on Patients’ Hospitalization

Sixteen isolates were recovered from various clinical specimens [four from blood, four from urine, three from bronchoalveolar lavage (BAL), three from pus, two from other types] and one from a screening culture (rectal swab) from 17 patients with either *P. aeruginosa* infection or colonization, hospitalized in hospitals H1 (15 patients) and H2 (two patients). The patients in H1 were hospitalized in both COVID-19 and non-COVID-19 ICUs and medical departments. The mean age of the patients was 68 years (range 32–90 years), and nine were female. Patients with bloodstream infections were treated with colistin. In five patients in hospital H1, the NDM-producing *P. aeruginosa* isolates were recovered from specimens sampled upon admission to the hospital from another healthcare facility or from patients with a history of previous hospitalizations. In the remaining patients, the NDM-producing *P. aeruginosa* isolates were recovered, on average, 47 days (range 10–220 days) from their admission to hospital H1. In hospital H2, the NDM-producing *P. aeruginosa* isolates were recovered 10 and 12 days after the two patients’ admissions (Table 1).

### 3.2. Antimicrobial Susceptibility Testing

All *P. aeruginosa* isolates exhibited high-level resistance to piperacillin/tazobactam, ceftazidime, cefepime, imipenem, meropenem, gentamicin, amikacin, and ciprofloxacin while being susceptible to aztreonam (except for 1 isolate with MIC > 64 mg/L) and colistin (MIC ≤ 2 mg/L, range 0.5–2 mg/L) (Appendix A).

### 3.3. Molecular Typing

MLST analysis characterized all but one of the isolates as ST773, which were associated with the same DLST 26–46, while the remaining isolate (Pae 9965), which was isolated from hospital H1, was typed as ST308, and it was associated with DLST 23–22 (Appendix A).

### 3.4. Detection of Antimicrobial Resistance Genes and Known Chromosomal Point Mutations

All *P. aeruginosa* isolates were found to be positive for *bla*_NDM_ and negative for *bla*_VIM_, *bla*_KPC_, and *bla*_OXA-48_ by conventional PCR assays. By the MDR flow cell assay, all isolates except one (*P. aeruginosa* 9965) were found to carry *bla*_NDM_ along with the methylase *rmtB*, which confers resistance to all aminoglycosides. Furthermore, all isolates were found to be positive for the *sul-1* gene, conferring resistance to sulfonamides, and the mutation T83I in the DNA gyrase subunit A of *gyrA* for resistance to quinolones. In addition, in the *P. aeruginosa* 9965 isolate, the *aac*(6’)-*Ib* and *qnrS* genes, conferring resistance to quinolones, were also detected.

A WGS-based resistome analysis of six isolates confirmed the aforementioned results and, furthermore, provided data for the entire resistome. A total of 14 ARGs were detected in five ST773 isolates, including ARGs conferring resistance to aminoglycosides [*aadA11*, *aph(3’)-IIb*, *and rmtB4*], β-lactams (*bla*_NDM-1_, *bla*
_PAO_*/bla*
_PDC-*16*_, and *bla*
_OXA-395_), chloramphenicol and its derivative florfenicol (*catB7* and *floR*), quinolones (*crpP* and *qnrVC1*), fosfomycin (*fosA*), sulfonamides (*sul1*), tetracycline [*tet(G)*], and the biocide quaternary ammonium (*qacE*).

In the remaining ST308 isolate, a total of 20 ARGs were detected, including ARGs for resistance to aminoglycosides [*aac(6’)-Ib*, *aac(6’)-II*, *aac(3)-Id*, *aph(3’)-IIb*, *aadA11*, and *rmtF2*], beta-lactams (*bla _NDM-1_*,*bla_PAC-1_*_,_
*bla*_PDC-19a_, *bla*_OXA-10_, and *bla*_OXA-488_), chloramphenicol (catB7), quinolones (*crpP* and *qnrVC1*), fosfomycin (*fosA*), bleomycin (*ble*), sulfonamides (*sul1*), trimethoprim (*dfrB*5), macrolides (*msr*E), and the biocide quaternary ammonium (*qacE*).

Additionally, the point mutations S87L in DNA topoisomerase IV subunit A of *par*C and the T83I in DNA gyrase subunit A of *gyrA*, conferring resistance to fluoroquinolones, were detected in all six isolates of both STs (Appendix A).

### 3.5. Genetic Environment of bla_NDM-1_

Hybrid assembly showed that, in ST773 *P. aeruginosa* strain 9912, *bla*_NDM-1_ was inserted into the *P. aeruginosa* chromosome. The *bla*_NDM-1_ gene was located within the integrative and conjugative element (ICE) ICE*6600-like*, as previously described in the *P. aeruginosa* strain P-600 (GenBank accession no. CP053917) isolated from South Korea [24].

In ST308 *P. aeruginosa* strain 9965, the *bla*_NDM-1_ gene was also localized in the *P. aeruginosa* chromosome; however, blastN analysis showed that *bla*_NDM-1_ was part of ICE*_Tn4371_6385*, as previously found in *P. aeruginosa* strain PASGNDM345 (GenBank accession no. CP020703) recovered from Singapore [25].

Additionally, sequence analysis with PlasmidFinder did not detect any plasmids in either isolate.

### 3.6. Detection of Virulence Genes

According to the virulence factor database (VFDB), all isolates from both ST773 and ST308 were found to harbor genes of 11 different VF classes, including genes for adherence (flagella, lipopolysaccharide, type IV pili), antimicrobial activity (phenazines biosynthesis), antiphagocytosis (alginate production), biosurfactant (rhamnolipid biosynthesis), enzyme production (phospholipase C and D), iron uptake (pyochelin and pyoverdin production), protease and elastase production, quorum sensing (*las* and *rhl* QS systems), regulation, secretion systems [type VI secretion system, type 3 secretion system (TTSS) with absence of *exoS* but presence of *exoT*, U, and Y genes], and toxins (exotoxin A, pyocyanin) (Appendix A).

### 3.7. Phylogenetics

The SNP analysis of the five ST773 isolates showed a cluster with a very close relationship between isolates 9912, 10071, 9953, and 9958 (0–6 SNPs) and an 11–15 SNP distance to isolate 9964. Notably, isolates 9912 from hospital H2 and 9953 from hospital H1 were closely related (0 SNPs), and both were related to 9958 (2 SNPs) from hospital H1, while isolate 10071 from hospital 2 differed by 4–6 SNPs.

Sample 9965 differed by >7800 SNPs from the other isolates. It was initially included in the phylogenomic analysis to exclude the possibility that the difference in the STs was a one- or few-allele-difference sequence type. And, thus, we did not include it in further SNP analyses.

## 4. Discussion

The emergence and dissemination of international high-risk *P. aeruginosa* clones worldwide poses both clinical and public health challenges. From a clinical perspective, these MDR and XDR clones, well adapted to the hospital environment, cause difficult-to-treat infections or, even worse, hospital-acquired infection outbreaks [19,20]. From a public health point of view, the remarkable ability of *P. aeruginosa* to acquire various resistance genes through horizontal transfer, driving the development of MDR strains, raises concerns about the risk factors, the identification of populations at risk, and the measures to be taken for the control of their further dissemination, including close monitoring by laboratory surveillance.

In the present study, we aimed to characterize emerging NDM-1-producing *P. aeruginosa* isolates recovered from patients hospitalized in two hospitals in the Attica region, the main metropolitan area of Greece, in 2022 in order to give insights into their resistome, their virulome, the genetic environment of *bla*_NDM-1_, and their molecular epidemiology. Notably, VIM-type MBL has been the main carbapenemase type in the *P. aeruginosa* bacterial population in the hospital setting in the country since the late 1990s [9,10,11,12,13], and thus, the emergence and further dissemination of another mechanism should be investigated from a laboratory surveillance perspective.

The study isolates had high-level resistance to all first-line antipseudomonal β-lactam antibiotics, as well as carbapenems, aminoglycosides, and fluoroquinolones, being susceptible only to colistin, at least among the antibiotics tested routinely in a clinical microbiology laboratory. This extensive resistance profile has severe clinical implications since it poses challenges to the selection of both an appropriate empirical and an effective targeted treatment and complicates the management of infected patients.

Furthermore, the isolates were found to harbor a wide variety of genetic determinants conferring resistance to the aforementioned classes of antibiotics and also to chloramphenicol, fosfomycin, sulfonamides, and tetracycline, leading to their XDR phenotype. The detection of *bla*_NDM-1_ and other antibiotic resistance genes (ARGs) in clinical settings has several implications for both patient care and healthcare practices. As for the treatment options for NDM-producing *P. aeruginosa* infections, both first-line anti-pseudomonal antibiotics and the newer combinations ceftolozane/tazobactam, ceftazidime/avibactam, meropenem/vaborbactam, and imipenem/relebactam should be avoided, limiting dramatically the treatment options to older and more toxic drugs like colistin or the newer cefiderocol and the combination of aztreonam with avibactam [24]. Furthermore, the simultaneous presence of the genes *rmtB* or *rmtF* for the respective methylases, which modify the cellular target of aminoglycosides, is further concerning because they confer resistance to all clinically available aminoglycosides. Apart from ARGs conferring low-level resistance to quinolones, the detected combination of mutations in *gyrA* T83 and *parC* S87, conferring high-level resistance to fluoroquinolones, renders also this antibiotic class ineffective for the treatment of infections with such isolates.

The majority of isolates from both hospitals were found to belong to the international high-risk clone ST773, recovered in H1 since early 2022 and in H2 since late 2022, while a unique isolate from H1 in mid 2022 was allocated to ST308. The detection of a plethora of ARGs in two high-risk *P. aeruginosa* clones also highlights the possibility of the co-selection and co-transfer of these ARGs in an environment of heavy antibiotic use. Furthermore, the absence of plasmids in the studied *P. aeruginosa* isolates and the identification of *bla*_NDM-1_ within integrative and conjugative elements (ICEs) indicate the ability of these isolates to integrate foreign genetic material into their chromosomes. Additionally, this finding suggests that the dissemination of *bla*_NDM-1_ could be mainly associated with the clonal spread of these two high-risk clones without excluding the possibility of the horizontal transfer of ICEs carrying *bla*_NDM-1_ to other important *P. aeruginosa* clones or even other species. Thus, as is clearly understood, the detection of international MDR/XDR high-risk clones in clinical settings has, apart from clinical implications, major epidemiological, infection control, and public health relevance and challenge [25].

The MLST of MDR *P. aeruginosa* isolates is currently the gold standard for the detection of international high-risk clones [20,25]. Since MLST is considered to be time-consuming and expensive, at least for routine clinical use, Double-Locus Sequencing Typing (DLST), a typing scheme first reported for *P. aeruginosa* in 2014, has been introduced, which uses the partial sequencing of two highly variable loci: ms172 and ms217 [23]. DLST has been previously implemented in clinical *P. aeruginosa* isolates from Greek hospitals [12]. Furthermore, the DLST scheme was compared to MLST in a number of clinical and environmental *P. aeruginosa* isolates, proving that when epidemiological and phylogenetic analyses are conducted at a local level, MLST can be replaced by DLST [26]. Moreover, DLST has been shown to provide satisfactory results for the detection of high-risk clones [26].

The MLST typing findings for ST773, along with the epidemiological data of patients’ hospitalization, are consistent with both possible inter- and intra-hospital transmission of this clone during the study period, mainly in hospital H1. Besides the close temporal–spatial linkage of the hospitalization periods among cases of infection or colonization with ST773 *P. aeruginosa* isolates in H1, in four patients, the specific ST773 isolate was recovered on the day of their admission to H1 from another hospital, indicating an unnoticed spread of NDM-1-producing *P. aeruginosa* isolates in Greek hospitals, at least in the metropolitan region of Attica.

With the advancement of new technologies, whole-genome sequencing (WGS) has been used in recent studies on *P. aeruginosa* epidemiological investigations in hospital settings [27], providing the ultimate discriminatory typing power. Accordingly, with the SNP analysis that we implemented in the five ST773 *P. aeruginosa* isolates (three from H1 and two from H2), we confirmed the aforementioned hypothesis, since the isolates were closely related to each other (range of 0–15 SNPs), indicating a clonal spread, both inter- and intra-hospital. This clonal spread emphasizes the need for strengthening infection control practices.

In addition, with the WGS analysis, a large number of genetic determinants of virulence factors were detected, both cell-associated and extracellular ones, in all six isolates. The identified virulence genes in the study isolates contribute significantly to their pathogenicity by enhancing adhesion, toxin production, biofilm formation, iron acquisition, and resistance to host defenses [1]. The presence of this wide variety of VFGs in the study isolates is indicative of the virulence potential of the *bla*_NDM-1_-positive ST773 and ST308 clones, which may further pose extra challenges for the recovery of critically ill patients in our hospitals. Notably, all isolates exhibited an *exoU*^+^ genotype in their TTSSs, indicative of high virulence potential in addition to their multidrug-resistant phenotype [1,28].

Regarding ST773, it has been characterized as a high-risk clone disseminated worldwide [19,20]. However, only as recently as 2018 was ST773 associated with *bla*_NDM-1_, as described in cases in Hungary [29] and the United States, the latter from a critically ill patient previously hospitalized in India [30], both in 2018. Notably, the genetic environment of *bla*_NDM-1_ in the ST773 *P. aeruginosa* isolate was further studied and reported as ICE*6660-like* [30] since it was found to be highly similar to ICE*6660* carrying the *bla*_NDM-1_ gene and the 16S rRNA methyltransferase gene *rmtD3* in a *P. aeruginosa* ST234 isolate from a sample collected earlier in Poland [31]. This ICE*6660-like*, carrying *aacC3*, *bla*_NDM-1_, and *rmtB4* genes, was found later in ST773 *P. aeruginosa* isolates from South Korea in 2019 [32] and Nepal in 2019–2020 [33], as well as in the present study in 2022. Notably, in 2022, the Netherlands [34] reported on the identification of four ST773 NDM-1-producing *P. aeruginosa* cases from Ukrainian refugees. Furthermore, since 2023 and early 2024, ST773 NDM-1-producing *P. aeruginosa* has been isolated in Canada [35], South Africa [36], Morocco [37], and Spain in patients transferred from Ukraine [38], indicating a successful, emerging metalloenzyme–high-risk clone combination.

Besides the predominant ST773 in the present study, ST308 was identified in one isolate from hospital H1 in July 2022. Notably, the respective patient was hospitalized for 23 days in the ICU of another hospital, and the *P. aeruginosa* isolate was recovered on the day of admission to hospital H1, indicating an ongoing latent circulation of ST308 as well in the hospital sector in the Attica metropolitan area and probably elsewhere. Indeed, one year later, in May–July 2023, a cluster of ST308 NDM-1-producing *P. aeruginosa* isolates was reported in a university hospital in central Greece [17]. The index patient was a young woman who had previously been hospitalized in the ICUs of various hospitals in Greece, indicating the scenario of a silent emergence and circulation of NDM-1 carbapenemase in the *P. aeruginosa* nosocomial population in the country.

ST308 has also been characterized as an epidemic high-risk clone disseminated worldwide [19]. Furthermore, it has been suggested that ST308’s intraclonal diversity is a driver of its adaptation and persistence in hospital settings and, therefore, its epidemic behavior [39]. *P. aeruginosa* ST308 has recently been associated with *bla*_NDM-1_, mainly in Southeast Asia, specifically Singapore [40,41,42] and Malaysia [43]. The genetic environment of *bla*_NDM-1_ in *P. aeruginosa* ST308 strains previously reported, as well as observed in our isolate, shows that *bla*_NDM-1_ was chromosomally inserted into the integrative and conjugative element ICE*_Tn4371_6385* and was part of one cassette along with two other resistance genes, *floR* and *msrE* [17,40], mediating the transferability and co-selection of carbapenem resistance. Furthermore, since previous studies have demonstrated the difficulties in eradicating *P. aeruginosa* ST308 in the hospital environment in spite of stringent infection control and cleaning measures, leading to sustained outbreaks, continued surveillance of ST308 *P. aeruginosa* is considered critical [39].

Our findings, in combination with the recent report from central Greece in 2023 [17], should be regarded along with the respective timing; the COVID-19 pandemic forced Greece, as many other countries worldwide, to slow down or temporarily discontinue or even postpone national plans and other initiatives to fight AMR, re-allocating both human and budget resources to cover COVID-19 public health emergencies and duties [44]. In this regard, similar to the ST308 NDM-producing *P. aeruginosa* outbreak in the ICU of the University Hospital in central Greece, the exact time and route of entrance and further transmission of ST773 NDM-producing *P. aeruginosa* in hospitals H1 and H2 remain unclear.

The lack of information on travel history and, thus, on a possible epidemiological link for the importation of *bla*_NDM-1_-mediated carbapenem resistance in the *P. aeruginosa* nosocomial population in Greece is one of the possible limitations of our study. Furthermore, novel antibiotics like cefiderocol and aztreonam/avibactam were not tested since they were not available. In addition, whole-genome sequencing was performed only in six isolates; however, we tried to include isolates from both hospitals and isolates from patients with intra- and inter-hospital transfer histories. Furthermore, for two isolates, in addition to short-read sequencing, we performed long-read sequencing in order to obtain better genome assemblies to thoroughly study the *bla*_NDM-1_ genetic environment in both ST773 and ST308 isolates.

To our knowledge, this is the first study to report on the genomic characteristics of ST773 NDM-1-producing *P. aeruginosa* in Greece isolated since the beginning of 2022 and even earlier. Furthermore, our study revealed a sporadic ST308 isolate during the same year, suggesting that two different events contributed to the emergence of NDM-1-producing *P. aeruginosa* isolates in Athenian hospitals in 2022. Our findings on the simultaneous emergence of two high-risk MDR/XDR *P. aeruginosa* clones with a wide array of virulence genes triggered a national multicenter genomic surveillance study under the auspices of the EURGen-RefLabCap project (https://www.eurgen-reflabcap.eu/, in 2023 with 17 participating tertiary hospitals all over the country, which is in the final analysis stage.

Indeed, intensifying genomic surveillance at the national and international levels to study and monitor the emergence, adaptation, and spread of high-risk CRPA clones is of utmost importance to understand the interplay between resistance and virulence in such a challenging pathogen for both clinical medicine and public health. However, the importance of the early detection of MDR/XDR high-risk clones at a local level to guide efficient antimicrobial treatment of severe infections and to prevent further transmission in hospital settings should also be emphasized.

## Figures and Tables

**Table 1 microorganisms-12-01753-t001:** Details on the isolation of the 17 studied NDM-1-producing *P. aeruginosa* isolates from patients hospitalized in hospitals H1 and H2, Attica region.

Patient No.	Age	Sex	Isolate No.	Hospital	Department of Hospitalization	Specimen	Date of Isolation	Days from Hospital Admission to Isolation
1	73	M	9953	H1	COVID-19 ICU	Blood	17 January2022	30
2	63	F	9959	H1	COVID-19 ICU	BAL	17 January 2022	11
3	59	M	9954	H1	COVID-19 ICU	Tissue	3 February 2022	23
4	67	F	9955	H1	ICU	Blood	2 March 2022	23
5	32	M	9956	H1	ICU	Wound	8 March 2022	220
6	64	F	9957	H1	Neurology	Blood	10 May 2022	43
7	70	F	9958	H1	Medical	Blood	17 May 2022	14
8	77	M	9951	H1	COVID-19	Urine	22 May 2022	0
9	51	F	9960	H1	Neurology	Rectal swab	7 June 2022	116
10	76	F	9962	H1	COVID-19	Drainage	8 July 2022	30
11	90	M	9963	H1	Medical	Urine	22 July 2022	0
12	53	M	9964	H1	Neurosurgery	BAL	23 July 2022	54 [Transferred from ICU to NS]
13	58	M	9965	H1	Cardiology	BAL	27 July 2022	0 [Transferred from ICU of another hospital]
14	76	M	9966	H1	Medical	Urine	18 August 2022	0
15	89	F	9952	H1	Medical	Urine	16 September 2022	0
16	88	F	9912	H2	Medical	pus	06 October 2022	10
17	67	F	10071	H2	Medical	pus	12 December 2022	12

## Data Availability

All the data associated with this manuscript are provided within the manuscript.

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
