# Peer review of "Emergence of NDM-1-Producing Pseudomonas aeruginosa Nosocomial Isolates in Attica Region of Greece"

_microorganisms, 2024, doi:10.3390/microorganisms12091753_

Round 1

Reviewer 1 Report

Comments and Suggestions for Authors

This work is interesting and well done.

However some minor revisions are necessary:

Table 1 and table 2 are not necessary, write in the text the results.

Species names in the text MUST be in Italic

These infections are acquired during the hospitalization, but have you find a cluster??

Have you identifed other bacteria with this gene??

The treatment of these infections could be interesting to know, with the aim to understand if the presence of this microorganism was opportunitic or not.

Comments on the Quality of English Language

Minor revision is necessary, only few mistakes throghout the entire manuscript, but not serious problem

Reviewer 2 Report

Comments and Suggestions for Authors

The manuscript " Emergence of NDM-1 producing Pseudomonas aeruginosa nosocomial isolates in Attica region of Greece" aims to report on the emergence and spread of NDM-1 producing multi-drug resistant P. aeruginosa in Attica, Greece, in 2022-, -detailing-their resistome and molecular epidemiology. Although the paper presents a certain scientific interest, there are some concerns regarding the data's validity and overall results. Here are some important comments:

1.      Describe how H2 isolates were identified and selected, with specific criteria for randomly selecting isolates from H1.

2.      For better understanding and comparison purposes, a table summarizing the MIC results is provided.

3.      Provide information concerning the validation and sensitivity/specificity of the MDR Direct Flow Chip assay in detecting AMR genes.

4.      List yield and purity measurements of your extracted DNA to ensure others can repeat your work.

5.      Why did you use DLST on four isolates? Also, tell us how you picked these four.

6.      You uploaded the sequence data to databases and gave their accession numbers so anyone can check them.

7.      Explain why you chose 280x average coverage for Illumina sequencing and how this depth covers the whole genome.

8.      Provide more details about how you filtered and fixed errors in the Nanopore sequencing data.

9. Explain why the Z-score cutoff at SNP spots and their minimum depth matter for the phylogenomic studies you picked.

10.   Explain why you chose the blaNDM-1 P. aeruginosa ST773 from South Korea as the reference genome and how it links to the isolates in this study.

11.   Share data on how you checked the INVIEW Genome by-product for consistency and repeatability.

12.   You picked isolate 9912 and isolate 9965 for extra Nanopore sequencing.

13.   Tell us about the amount of HMW DNA you got and how you checked its quality.

14.   Justification of the choice of SPAdes v3.11.1 for de novo assembly and its comparison to other tools in this role.

15.   Mention the criteria followed while selecting Flye 2.9.3 for de novo assembly and Medaka 1.7.2 for assembly polishing.

16.   Performance details and accuracy of ABRicate, AMRFinderPlus, VFDB, ICEfinder, and PlasmidFinder in detecting ARGs and virulence factors.

17.   The implications of high-order resistance patterns of the P. aeruginosa isolates, especially piperacillin-tazobactam, ceftazidime, and cefepime.

18.   Implications of the detection of blaNDM, the methylase rmtB, and other ARGs in clinical settings.

19.   Comment on the possible clinical consequences of detecting 14 ARGs in ST773 isolates and 20 ARGs in the ST308 isolate.

20.   Implication of the identification of blaNDM1 within ICE elements in ST773 and ST308 isolates.

21.   The absence of plasmids detected in both isolates and its impact on the dissemination potential of blaNDM-1.

22.   Epitomize what clinical relevance the identified virulence genes contribute to across different VF classes.

23.   Based on available evidence, explain what contribution virulence factors like exotoxin A, pyocyanin, and T3SS genes could make to patient outcomes.

24.   Describe in more detail the phylogenetic relationship and SNP analysis results.

25.   The genetic relatedness of the four isolates, 9912, 10071, 9953, and 9958, has clinical and epidemiological implications.

26.   Reasons for excluding isolate 9965 from further SNP analysis because of a significant genetic divergence.

Comments on the Quality of English Language

Some minor editing of the English language is necessary for clarity and precision.

Round 2

Reviewer 2 Report

Comments and Suggestions for Authors

After thoroughly reviewing the revised manuscript and considering the authors' revisions and responses to the referee's comments, I find that the manuscript has been significantly improved. The authors have effectively addressed the concerns, enhancing their study's clarity and scientific rigor. The revisions have clarified the methodology, improved the presentation of results, and strengthened the discussion and conclusions.

Therefore, I believe that the manuscript now meets the standards required for publication in Microorganisms and recommend that it be accepted for publication.

Thank you for considering my recommendation.